# The Efficacy of Thrombin-Gelatin Matrix in Hemostasis for Large Breast Tumor after Vacuum-Assisted Breast Biopsy

**DOI:** 10.3390/jpm12020301

**Published:** 2022-02-17

**Authors:** Yen-Dun Tony Tzeng, Shiuh-Inn Liu, Being-Whey Wang, Yu-Chia Chen, Po-Ming Chang, I-Shu Chen, Jim Jinn-Chyuan Sheu, Jui-Hu Hsiao

**Affiliations:** 1Department of Surgery, Kaohsiung Veterans General Hospital, Kaohsiung City 813, Taiwan; seeoutony@gmail.com (Y.-D.T.T.); siliu@vghks.gov.tw (S.-I.L.); bwwang@vghks.gov.tw (B.-W.W.); ycchen@vghks.gov.tw (Y.-C.C.); pmchang@vghks.gov.tw (P.-M.C.); ischen@vghks.gov.tw (I.-S.C.); 2Institute of Biomedical Sciences, National Sun Yat-sen University, Kaohsiung City 804, Taiwan; sheu.jim@gmail.com; 3Department of Surgery, Kaohsiung Municipal Minsheng Hospital, Kaohsiung City 802, Taiwan

**Keywords:** large breast tumor, vacuum assisted breast biopsy, thrombin-gelatin matrix

## Abstract

Background: Vacuum-assisted breast biopsy (VABB) for benign breast tumor excision is a developing trend in breast surgery. The most common complication of VABB is hematoma. We assessed the efficiency of the thrombin-gelatin matrix (TGM) for hemostasis after VABB. Methods: From December 2013 to June 2017, 147 patients with breast tumors > 2 cm in size were treated with a 7-gauge ultrasound-guided EnCor EnSpire^®^ breast biopsy system. After VABB, the TGM was applied using an iron-tube device. After injection, brief external compression for 15 min and postoperative bandage compression for approximately 12 h were applied. The medical records were reviewed and analyzed for hematoma and acute bleeding at 1 and 3 months after VABB. Results: A total of 72 patients received hemostasis via TGM, and 75 patients received hemostasis by compression. The rates of postoperative acute bleeding in the TGM group were significantly lower than those in the non-TGM group (5.5% vs. 22.7%, *p* = 0.003). Among patients with hematoma, there was no statistically significant difference between the two groups (25% vs. 26.7%, *p* = 0.85). Conclusions: This is the first cohort study to apply the TGM hemostatic matrix for post-VABB hemostasis. The TGM hemostatic matrix could be an option for patients with large breast tumors.

## 1. Introduction

Vacuum-assisted breast biopsy (VABB) for benign breast tumor excision is a developing trend in breast surgery. VABB can retrieve a larger volume of breast tissue than a core needle biopsy. A previous study reported that the false negative rate was only 0.1% among 1512 VABB cases [1]. Thereby, VABB can contribute reliable pathological results.

Although VABB is feasible in clinical practice, complications are inevitable. The most common complication of VABB is hematoma. Other complications, such as postoperative pain, subcutaneous bleeding, skin defects, and pneumothorax, have also been reported [2,3]. The most common procedure for achieving hemostasis in VABB is compression with pressure. Suturing of bleeders are rarely performed during VABB.

Recently, numerous products have been introduced to achieve hemostasis in different ways, such as the thrombin-gelatin matrix (TGM), topical hemostatic agents (HA) (e.g., sponges), thrombin, fibrin glue, and other types of surgical sealants. Several studies have reported the use of TGM compared with other hemostatic agents. TGM has been demonstrated to be an efficacious method to reduce the time to achieve hemostasis and the length of hospital stay, resulting in less consumption of health resources [4,5,6].

The TGM is a hemostatic matrix composed of a bovine gelatin matrix (FLOSEAL, Baxter Healthcare Corporation, Deerfield, IL, USA). TGM has been used in multiple surgical fields, such as cardiac and vascular surgeries, orthopedic surgery, tonsillectomy and adenoidectomy, sinus surgery, thyroidectomy, gynecologic surgery, urologic procedures, and lacrimal surgery [6].

The postoperative hematoma-associated factors of VABB are lesion size, number of lesions, and efficacy time of bandage [7,8]. We would like to reduce the percentage of hematoma by adding hemostatic agents during surgery. This is the first study to demonstrate the use of TGM in breast surgery for VABB hemostasis.

## 2. Materials and Methods

### 2.1. Patients

We retrospectively reviewed 147 consecutive patients who underwent VABB for breast tumor excision at Kaohsiung Veterans General Hospital from December 2013 to June 2017. All patients underwent a preoperative breast sonography and the reports were either breast imaging-reporting and data system (BI-RADS) category 2 or BI-RADS category 3. Some of patients received TGM injection intraoperatively under patient request, for hemostatic reasons. Patients without TGM injection received external compression for 15 min and postoperative bandage compression for approximately 12 h. After VABBs were performed, all tissues were examined by pathologists with formal pathological reports.

VABB was performed using the EnCor Enspire breast biopsy system (BARD, Murray Hill, NJ, USA), and a 7-gauge biopsy needle for each VABB was selected. A frequency of 10 MHz ultrasound was required during the operation. After intravenous general anesthesia, a 7-gauge probe was inserted into the breast through a 3-mm skin incision. Under direct ultrasound visualization, the probe was guided under the tumor as the initial biopsy position. Multiple core samples were taken until the tumor was completely removed, as determined by real-time ultrasound imaging. After VABB, TGM was applied via an iron-tube device for injection into the post-biopsy space (Figure 1). Brief external compression for 15 min before final dressing was performed. All procedures were performed by the same surgeon. All patients were enrolled by the Kaohsiung Veterans General Hospital Institutional Review Board (VGHKS17-CT6-03) and Kaohsiung Veterans General Hospital Cancer Database.

### 2.2. Exclusion

Breast masses < 2 cm were excluded. Patients who were taking anticoagulants and had bleeding tendencies were excluded. Patients with liver cirrhosis and chronic renal failure were excluded. One male was also excluded.

Ultrasound was performed 24 h after surgery. We defined hematoma as a diameter of > 1 cm. Follow-up was performed at 1, 3, and 6 months after VABB. Acute bleeding was defined as persistent bleeding from the insertion wound at 24 h after VABB.

### 2.3. Statistical Analysis

The statistical strategy was analyzed using SPSS version 25. Continuous data were analyzed using the Mann–Whitney U test. Categorical variables were analyzed using Fisher’s exact test. Binary logistic regression analysis was performed for univariate analysis of factors for hematoma and acute bleeding. *p* value < 0.05 was defined to indicate significant differences.

## 3. Results

Among the 147 patients, a total of 206 breast tumors were removed by 7-gauge ultrasound-guided VABB. The median age of the patients was 38 years (Table 1). The histopathological reports of the 206 lesions revealed that four lesions were malignant (1.9%): two invasive ductal carcinomas and two intraductal carcinomas. The 202 benign lesions were as follows: 114 fibroadenomas (56.4%), 40 fibrocystic lesions (19.8%), 12 benign phyllodes tumors (5.9%), 9 intraductal papilloma (4.4%), 6 usual ductal hyperplasia (3.0%), 6 atypical ductal hyperplasia (3.0%), 7 harmatoma (3.4%), 2 granulomatous mastitis (1.0%), 2 lactating mastitis (1.0%), 2 xanthogranulomatous inflammation (1.0%), and 2 pseudoangiomatous stromal hyperplasias (1.0%). Both groups had two patients with malignancy at the final pathological report. Patients diagnosed with breast malignancy underwent a standard lumpectomy or mastectomy with sentinel lymph node biopsy as definite oncological surgery. No mortality was noted after VABBs.

A total of 72 patients received hemostasis via TGM and 75 patients received hemostasis by compression. There were four patients (5.5%) with bleeding (Figure 2) and 18 (25%) with hematoma in the TGM group. However, 17 patients (22.7%) had bleeding and 20 (26.7%) had hematoma in the non-TGM group. The rates of postoperative acute bleeding in the TGM group were significantly lower than those in the non-TGM group (5.5% vs. 22.7%, *p* = 0.003). Among patients with hematoma, there was no statistically significant difference between the two groups (25% vs. 26.7%, *p* = 0.85) (Table 2).

Univariable analyses in hematoma group showed that there were no significant association with lesion number (*p* = 0.802), lesion size (*p* = 0.518) or application of TGM (*p* = 0.787) (Table 3). However, in the acute bleeding group, although there were no significant associations with lesion number (*p* = 0.434), or lesion size (*p* = 0.364), the application of TGM was affected independently (*p* = 0.005) (Table 4). All of hematoma were resolved during the 6-month follow-up.

## 4. Discussion

Jung et al. showed that ultrasound-guided core needle biopsy of a benign breast mass measuring 2 cm is sufficient to rule out malignancy, with an accuracy rate of 98.6% [9]. The advantage of VABB is that it provides not only the diagnosis but also the removal of the mass [10]. The most common complication after VABB is hematoma [11]. VABB is a minimally invasive surgery; therefore, neither electrical coagulation nor internal sutures for hemostasis are available. External compression can resolve most of the complications. Fu et al. introduced the effectiveness and safety of using a Foley catheter in VABB to prevent hematoma and bleeding [11,12], which was found to be less time consuming and to result in less bleeding and post-interventional hematoma.

The hematoma and acute bleeding rates were 25.8% (38/147) and 14.2% (21/147), respectively. Our result was similar to that reported by Fu et al. [12]. Schaefer et al. reported significantly more hematomas and acute bleedings for the 8-gauge-Mammotome^®^-system vs. the 11-gauge-Mammotome^®^-system (35.5% vs. 16.7%, *p* = 0.029; 41.9% vs. 8.4%, *p* < 0.001) [13]. Zheng et al. reported a hematoma rate of 11.4% but lacked acute bleeding rate. Zheng et al. also reported that the lesion size and number of lesions were independently associated with hematoma occurrence [7].

Removing larger lesions may lead to increased surgical space and the risk of vessel injury. It is difficult to treat refractory bleeding by external compression of the wound. Therefore, we filled the residual surgical space with TGM injection. Lesion number and size were not associated with hematoma or acute bleeding. However, our result showed that the application of TGM would decrease the rate of acute bleeding independently. 

The application of TGM in breast surgery has rarely been reported. Henkel et al. reported the presentation of breast pseudo-microcalcification on mammogram after TGM injection [14]. During our series, postoperative mammograms were performed in 10 patients, and no appearance of pseudo-microcalcification was noted. We suggest yearly to follow-up of the cases due to the possibility of breast pseudo-microcalcification. 

In our study, the malignancy rate after VABB was 1.9%. Lee et al. reported a malignancy rate of 5.4% after a 10-year VABB follow-up [1]. In order to expand the indications of VABB, there are several studies presenting VABB for breast cancer excision or biopsy confirmation for post-neoadjuvant chemotherapy [3,15,16]. However, a higher residual tumor rate in early breast cancer patients receiving VABB has been reported [17]. Therefore, we suggest that VABB is a better indicator of benign breast lesions, such as fibroadenoma, hematoma, lipoma, and benign papilloma. If malignancy is present after VABB, standard oncological surgery is recommended.

Recently, TGM has been applied into stereotactic brain biopsies in the management of intractable hemorrhage [18]. De Quintana-Schmidt et al. reported that TGM injection is a simple, safe, and effective stereotactic practice for managing persistent surgical bed bleeding that cannot be arrested by standard, conventional hemostatic methods. The method used in this study is very similar to our study, in which TGM was injected into the small tract after the biopsy. Both studies revealed a high efficiency of hemostasis after the TGM injection.

Walder et al. reported that TGM is cost effective due to its ability to reduce the likelihood of lymphocele formation. In a cost analysis by Euro currency, the mean cost per patient in the TGM group was €327 compared with the non-TGM cost per patient of €553 [19]. The reduction of operative blood loss would reduce the need for blood transfusions and hospital stay, and thus greatly reduce costs.

The limitations of this study are its sample size and it being a retrospective cohort study. These factors may result in bias with a characteristic heterogeneity. Prospective randomized control trials should be conducted in the future.

## 5. Conclusions

This is the first cohort study to apply TGM injection to ensure post-VABB hemostasis. Improved hemostasis during acute bleeding and a trend to prevent hematoma were observed after the TGM injection. In conclusion, TGM could be an alternative method to achieve better post-VABB hemostasis.

## Figures and Tables

**Figure 1 jpm-12-00301-f001:**
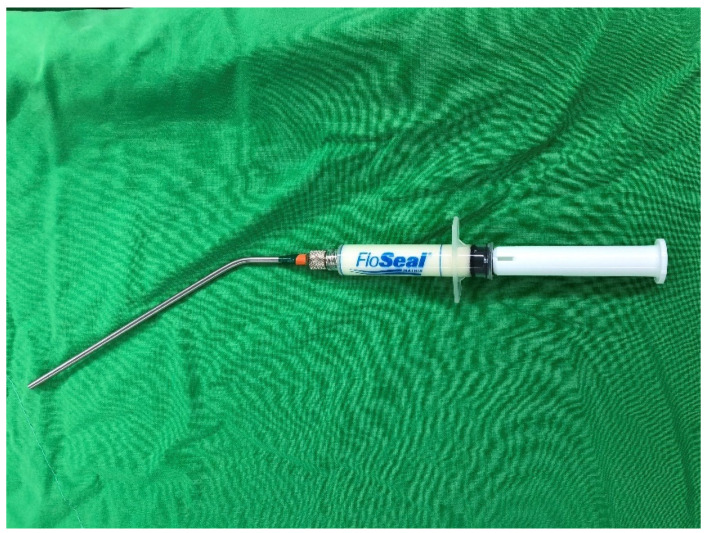
Iron-tube device for injection.

**Figure 2 jpm-12-00301-f002:**
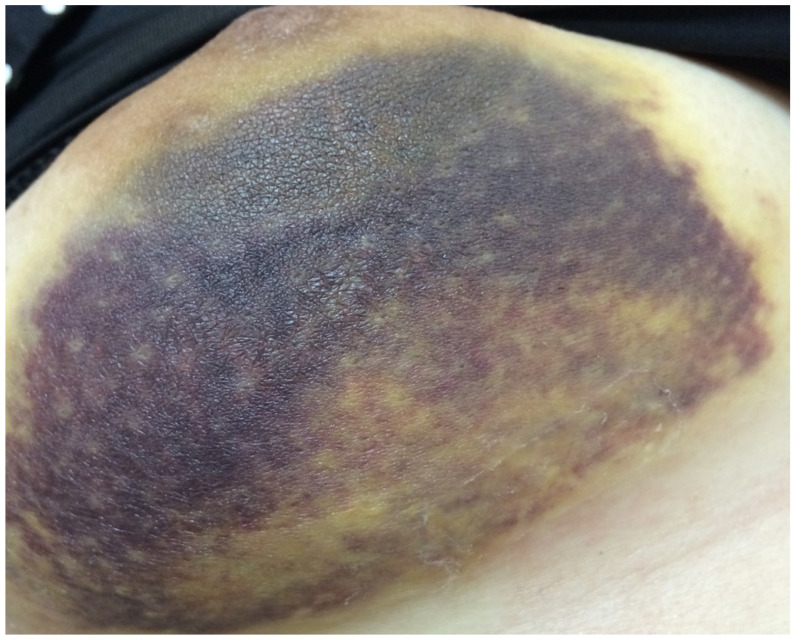
Patient with acute bleeding in Day 1.

**Table 1 jpm-12-00301-t001:** Patient demographics and lesion characteristics.

Characteristics	All Patients (*n* = 147)	All Lesion (*n* = 206)	Hemostasis with TGM (*n* = 72)	Hemostasis without TGM (*n* = 75)	*p* Value
Age			39 (17–78)	38 (14–68)	0.6
Size (cm)			2 (2–5)	2 (2–4.6)	0.42
Lesion number					1.0
Single	102 (69.4%)		50 (69.4%)	52 (69.3%)	
Multiple	45 (30.6%)		22 (30.6%)	23 (30.7%)	
Pathology					1.0
Benign		202 (98.1%)	100	102	
Malignant		4 (1.9%)	2	2	

TGM = Thrombin-gelatin matrix.

**Table 2 jpm-12-00301-t002:** Hematoma and acute bleeding.

Complications	Hemostasis with TGM (*n* = 72)	Hemostasis without TGM (*n* = 75)	*p* Value
Post-VABB hematoma			0.85
Yes	18 (25.0%)	20 (26.7%)	
No	54 (75.0%)	55 (73.3%)	
Acute bleeding			0.003
Yes	4 (5.5%)	17 (22.7%)	
No	68 (94.4%)	58 (77.3%)	

TGM = thrombin-gelatin matrix.

**Table 3 jpm-12-00301-t003:** Result of factors affecting Hematoma.

Parameters	Odds Ratio	95% CI	*p* Value
Lesion number (single/>2)	1.109	0.493–2.497	0.802
Lesion size (2/> 2 cm)	0.783	0.373–1.644	0.518
Thrombin-gelatin matrix (No/Yes)	1.108	0.528–2.327	0.787

CI = confidence interval.

**Table 4 jpm-12-00301-t004:** Result of factors affecting acute bleeding.

Parameters	Odds Ratio	95% CI	*p* Value
Lesion number (single/>2)	1.557	0.514–4.711	0.434
Lesion size (2/> 2 cm)	0.64	0.244–1.678	0.364
Thrombin-gelatin matrix (No/Yes)	5.225	1.648–16.57	0.005

CI= confidence interval.

## Data Availability

The data in this study are available from the corresponding author.

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
