# Peer review of "The Efficacy of Thrombin-Gelatin Matrix in Hemostasis for Large Breast Tumor after Vacuum-Assisted Breast Biopsy"

_jpm, 2022, doi:10.3390/jpm12020301_

Round 1

Reviewer 1 Report

Dear editor:

Thank you for inviting me to evaluate this article titled “The Efficacy of Thrombin-Gelatin Matrix in Hemostasis for Large Breast Tumor after Vacuum-Assisted Breast Biopsy”. In this study, the authors assessed the efficiency of the thrombin-gelatin matrix (TGM) for hemostasis after VABB, and found that TGM injection could reduce the probability of acute bleeding compared with postoperative bandage compression. The work presented in this paper is practical and logical. However, there are some problems that need further improvement as well:

  1. In the Discussion section, the authors only repeated the VABB and TGM background as they narrate in the Introduction section, and summarized their experiments, but lacked insights into their work. This can be shown in the following aspects:
    1. The authors need to compare their work with other published methods mentioned in Line 41 to Line 42 to present the pros and cons of TGM injection.
    2. It is noted that TGM is only effective in reducing acute bleeding compared to postoperative bandage compression. The authors should also analyze the cost-effectiveness of TGM. If the cost of TGM is much higher than the cost of postoperative bandage compression, TGM may not be satisfactory for all patients.
  2. The Introduction section is too general:
    1. [line 33] “A series of reports have indicated that VABB is a safe and efficient method for breast tumor excision and biopsy.” What is the concept or extent of “safe” and “efficient”? Since VABB has so many complications in terms of lines 35~39, why is VABB safe?
    2. [line 45] “TGM has been demonstrated to be efficacious in multiple surgical fields …”. This is similar to the one I listed in the comment 2a. It is important to note the efficiency of TGM applied in these fields.
    3. [line 31] “VABB can retrieve a larger volume of breast tissue than a core needle biopsy and can contribute to more reliable pathological results.” and [line 41] “such as topical hemostatic agents (HA) (e.g., sponges), thrombin, gelatin- thrombin, fibrin glue, and other types of surgical sealants.” Citations are needed to support their conclusions.

Reviewer 2 Report

I do not have any significant comments – I would like to thank the authors for the original and useful study, although performed with some limitations, as admitted by the authors.

From my point of view, the article can be published in the current form.

Author Response

Thank the editor and reviewer for the time they spent reviewing your manuscript.